# MODEL OBFUSCATION FOR SECURING DEPLOYED NEURAL NETWORKS

## ABSTRACT

More and more edge devices and mobile apps are leveraging deep learning (DL) capabilities. Deploying such models on devices – referred to as on-device models – rather than as remote cloud-hosted services, has gained popularity as it avoids transmitting user's data off of the device and for high response time. However, on-device models can be easily attacked, as they can be accessed by unpacking corresponding apps and the model is fully exposed to attackers. Recent studies show that adversaries can easily generate white-box-like attacks for an on-device model or even inverse its training data. To protect on-device models from white-box attacks, we propose a novel technique called *model obfuscation*. Specifically, model obfuscation hides and obfuscates the key information – structure, parameters and attributes – of models by *renaming*, *parameter encapsulation*, *neural structure obfuscation*, *shortcut injection*, and *extra layer injection*. We have developed a prototype tool *ModelObfuscator* to automatically obfuscate on-device TFLite models. Our experiments show that this proposed approach can dramatically improve model security by significantly increasing the difficulty of extracting models' inner information, without increasing the latency of DL models. Our proposed on-device model obfuscation has the potential to be a fundamental technique for on-device model deployment. Our prototype tool is publicly available at https://github.com/AnonymousAuthor000/Code2536.

## 1 INTRODUCTION

Numerous edge and mobile devices are leveraging deep learning (DL) capabilities. Though DL models can be deployed on a cloud platform, data transmission between mobile devices and the cloud may compromise user privacy and suffer from severe latency and throughput issues. To achieve high-level security, users' personal data should not be sent outside the device. To achieve high throughput and short response time, especially for a large number of devices, on-device DL models are needed. The capabilities of newer mobile devices and some edge devices keep increasing, with more powerful systems on a chip (SoCs) and a large amount of memory, making them suitable for running on-device models. Indeed, many intelligent applications have already been deployed on devices (Xu et al., 2019) and benefited millions of users.

Unfortunately, it has been shown that on-device DL models can be easily extracted. Then, the extracted model can be used to produce many kinds of attacks, such as adversarial attacks, membership inference attacks, model inversion attacks, *etc*. (Szegedy et al., 2013; Chen et al., 2017b; Shokri et al., 2017; Fang et al., 2020). The deployed DL model can be extracted by three kinds of attacks: (1) extracting the model's weights through queries (Tramèr et al., 2016). (2) extracting the entire model from devices using software analysis (Vallée-Rai et al., 2010) or reverse engineering (Li et al., 2021b). (3) extracting the model's architecture by side-channel attacks(Li et al., 2021a).

According to our observation, existing defense methods can be categorized into two different levels: (1) algorithm level, and (2) side-channel level. For securing the AI model at the algorithm level, some studies (Orekondy et al., 2019b; Kariyappa & Qureshi, 2020; Mazeika et al., 2022) propose methods to degenerate the effectiveness of query-based model extraction. While other studies (Xu et al., 2018; Szentannai et al., 2019; 2020) propose methods to train a simulating model, which has similar performance to the original model but is more resilient to extraction attacks. For securing

the AI model at the side-channel level, a recent work modifies the CPU and Memory costs to resist the model extraction attacks (Li et al., 2021a).

Although many attacks have been proposed to extract DL models, it is hard for adversaries to precisely reconstruct DL models that are identical to the original ones using queries or side-channel information. These attack cannot access the inner information of the model, which means they are black-box attacks. In contrast, since on-device models are delivered in mobile apps and hosted on mobile devices, adversaries can easily unpack the mobile apps to extract the original models for exploitation. It will enable serious intellectual property leakage and adversaries can further generate white-box attacks, which are much more effective than black-box attacks (Zhang et al., 2022). Despite that model extraction using software analysis may lead to severe consequences, to the best of our knowledge, the community has not yet been aware of this attack, and no effective defense method has been proposed against it.

In this paper, **we propose a novel model protection approach based on model obfuscation, which focuses on improving AI safety for resisting model extraction using software analysis**. Given a trained model and its underlying DL library (*e.g.*, PyTorch, TensorFlow, TFLite and etc.), an end2end prototype tool, *ModelObfuscator*, is developed to generate the obfuscated on-device model and the corresponding DL library. Specifically, *ModelObfuscator* first extracts the information of the target model and locates the source code in the library used by its layers. Then, it obfuscates the information of the models and builds a customized DL library that is compatible with the obfuscated model. To achieve this, we design five obfuscation methods, including: (1) *renaming*, (2) *parameter encapsulation*, (3) *neural structure obfuscation*, (4) *random shortcut injection*, and (5) *random extra layer injection*. These obfuscation methods significantly increase the difficulty of parsing the information of the model. The model obfuscation can prevent adversaries from reconstructing the model. In addition, adversaries also hard to transfer the trained weights and structure of models to steal intellectual property using model conversion because the connection between the obfuscated information and the original one is randomly generated. Experiments on 10 different models show that *ModelObfuscator* can against state-of-the-art model parsing and attack tools with a negligible time overhead and 20% storage overhead. Our contributions in this work include:

- We propose the model obfuscation framework to hide the key information of deployed DL models at the software level. It can prevent adversaries from generating white-box attacks and stealing the knowledge of on-device models.

- We design five obfuscation strategies for protecting on-device models and provide an end2end prototype tool, *ModelObfuscator*. This tool automatically obfuscates the model and builds a compatible DL software library. The tool is open-source available.

- We provide a taxonomy and comparison of different obfuscation methods in terms of effectiveness and overhead, to guide model owners in choosing appropriate defense strategies.

## 2 RELATED WORK

**Model Extraction Attacks and Defenses** For model extraction attacks, adversaries can effectively extract the model in black-box setting. They can use collected samples to query the target model to reconstruct the target model (Tramèr et al., 2016; Papernot et al., 2017; Orekondy et al., 2019a; He et al., 2021; Rakin et al., 2022), or use the synthetic sample to steal the information of target models (Zhou et al., 2020; Kariyappa et al., 2021; Yuan et al., 2022; Sanyal et al., 2022). For defending against the model extraction attacks, various methods (Orekondy et al., 2019b; Kariyappa & Qureshi, 2020; Mazeika et al., 2022) have been proposed to degenerate the performance of model extraction attacks. Some methods (Szentannai et al., 2019; 2020) have been proposed to train a simulating model, which has similar performance to the original model, but can reduce the effectiveness of attacks. In addition, watermarking is also a promising method to defend against the model extraction (Yang et al., 2019; Fan et al., 2019; Lukas et al., 2019).

**Adversarial Machine Learning:** Currently, adversaries can use many kinds of attacks to challenge the reliability of DL models, such as adversarial attacks, membership inference attacks, model stealing attacks, and model inversion attacks. For the adversarial attack, depending on the knowledge required by the adversary, adversarial attacks can be categorized into white-box attacks such as gradient-based attacks (Croce & Hein, 2020; Goodfellow et al., 2015; Kurakin et al., 2017; Papernot et al., 2016; Moosavi-Dezfooli et al., 2016; Madry et al., 2018; Moosavi-Dezfooli et al., 2017), and

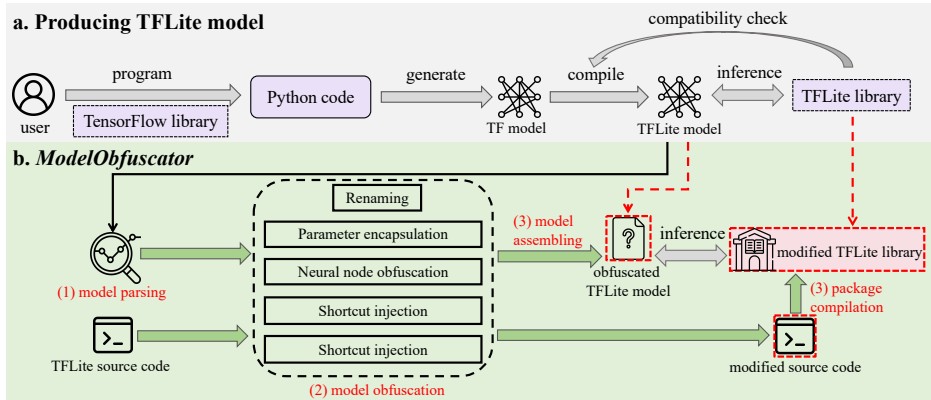

Figure 1: The working process of our *ModelObfuscator* on-device DL model obfuscation tool.

black-box attacks such as query-based attacks (Chen et al., 2017a; Ilyas et al., 2018a;b; Guo et al., 2019; Brendel et al., 2017; Cheng et al., 2018; Chen et al., 2020; Mopuri et al., 2018). For membership inference attacks, several studies (Shokri et al., 2017; Truex et al., 2019; Choquette-Choo et al., 2021; Carlini et al., 2022) challenge the privacy-preserving ability of model by predicting whether a sample is in the collected training set.

**Code obfuscation:** Code obfuscation methods are initially developed for hiding the functionality of the malware. Then, the software industry also uses it against reverse engineering (Schrittwieser et al., 2016). They provide complex obfuscating algorithms for programs like JAVA code (Collberg et al., 1997; 1998), including robust methods for high-level languages (Wang, 2001) and machine code level (Wroblewski, 2002) obfuscation. Code obfuscation is a well-developed technique to secure the source code. However, traditional code obfuscation approaches are not capable of protecting on-device models, especially for protect the structure of the models and their parameters. In this work, inspired by traditional code obfuscation, we propose a novel model obfuscation approach to obfuscate the model files and then produce a corresponding DL library for them.

## 3 METHODOLOGY

**Threat Model** The on-device model is usually saved as a separate file (e.g., .tflite file) and packed into the app package. The attackers can either download the target app from the app markets (e.g., Google Play and iOS App store) or extract the app package file (e.g., APK file for Android, and IPA file for iOS) from the hosting devices. These app package files can then be decompiled by off-the-shelf reverse-engineering tools (e.g., Apktool [1] and IDA Pro [2]) to get the original DL model file. Although many on-device DL platforms do not support some advanced functions like backpropagation, attackers can assemble the model architecture and weights into a differentiable model format, or they can use software analysis methods to generate attacks for the target model file (Li et al., 2021b; Huang & Chen, 2022). In this study, we will obfuscate the information of the model to disable the software analysis and model conversion tools, and generate a compatible DL library for the obfuscated model that only supports the forward inference function.

We chose the TensorFlow Lite (TFLite) platform to demonstrate our model obfuscation approach. TFLite is currently the most commonly used on-device DL platform. The steps to produce TFLite models are shown in the top half of Figure 1. Usually, DL developers use TensorFlow application programming interfaces (APIs) to define and train the TensorFlow (TF) model. The trained TF model is then compiled to a TFLite model. Note that TFLite has different implementations with TF, their operators (*i.e.*, layers) may not be compatible. Therefore, the TFLite library will check the compatibility during compilation. Once the compatibility check passed, the compiled TFLite model can run on devices using the TFLite library. Hence, the problem of obfuscating TFLite models is to design obfuscation strategies and make the obfuscated model compatible with the TFLite library.

We analyzed the current mainstream DL platforms (*i.e.*, TensorFlow and PyTorch) and identified the following two main findings. First, these platforms are open-source and provide a set of tools to build

---

[1] https://ibotpeaches.github.io/Apktool/
[2] https://hex-rays.com/ida-pro/

the library (*e.g.*, TFLite library) from the source code. Second, they officially support customized operators (*e.g.*, neural layers). Specifically, on top of these DL platforms, users could implement customized layers in C/C++ and compile customized layers to executing files (`.so` file in TensorFlow). Then, users can use these customized layers via high-level Python interfaces. Those features enable us to design obfuscation techniques to obfuscate model and DL library code together. The bottom half of Figure 1 shows the overview of our model obfuscation framework. Specifically, *ModelObfuscator* has three main steps: (1) model parsing, (2) model obfuscation, and (3) model assembling & library recompilation. In the following subsections, we detail our proposed *ModelObfuscator*.

## 3.1 MODEL PARSING

The first step of *ModelObfuscator* model obfuscation is to parse the deployed model to extract its key information. *ModelObfuscator* first extracts the structure information of each layer, including the name of layers (*e.g.*, `Conv2D`) and model structures (including model's input, output, layer ID and *etc.*). The extracted structure information are depicted using a data structure as shown in the right example. Then, *ModelObfuscator* will extract the parameter of each layer. Moreover, *ModelObfuscator* will identify the source code used by each layer by referring to the underlying libraries. The identified source code includes relevant packages or functions of the TFLite layers.

---

**Minimal model structure**

```
{
    "Layer name": "Conv2D",
    "Input": [0],
    "Output": [1],
    "ID": 0,
}
. . .
```

---

## 3.2 MODEL OBFUSCATION

After getting the model information and corresponding source codes, *ModelObfuscator* will obfuscate the model as well as the source codes. *ModelObfuscator* uses five obfuscation strategies: *renaming*, *parameter encapsulation*, *neural structure obfuscation*, *shortcut injection*, and *extra layer injection*.

***Renaming*** The most straightforward obfuscation strategy is the renaming of a layer. Usually, the layer's name contains important information, which is the function of this layer. For instance, "Conv2D" indicates a 2D convolution layer. Such information is useful for adversaries to reconstruct the model to generate white-box attacks or to obtain a similar surrogate model to conduct effective black-box attacks. To hide such important information, we randomly change each model layer's name. On the right is an example of an obfuscated `Conv2D` layer. *ModelObfuscator* automatically replaces the real name with the random meaningless string `Tbuszp`. Meanwhile, *ModelObfuscator* creates a copy of `Conv2D`'s source code and replaces the layer name (*i.e.*, `Conv2D`) in the source code with `Tbuszp`. Note that we modify the duplicate of `Conv2D` in case the modification affects other parts of the TFLite library. After adding modified source codes into the TFLite project, the recompiled TFLite library will recognize obfuscated layers as custom layers and correctly execute them at runtime.

---

**Obfuscated minimal model structure**

```
{
    "Layer name": "Tbuszp",
    "Input": [0],
    "Output": [1],
    "ID": 0,
}
. . .
```

---

**Source code registration (C/C++ code)**

TfLiteRegistration* Register_Conv2D()

↓

TfLiteRegistration* Register_Tbuszp()

---

***Parameter encapsulation*** Existing TFLite models have two main assets: model structure and parameters. The parameter can be obtained in the training period. Given an input, a TFLite model will compute the results using the input tensor and parameters that are stored in the model file. As we discussed above, parameter exposure is very dangerous. An adversary could use the parameter information to perform many kinds of white-box attacks, *e.g.*, adversarial attacks and model inversion attacks. Besides, an adversary can guess the function of this layer according to the shape of the parameter, because different layers have different numbers of parameters (*e.g.*, two in the convolution layer) and different shapes of parameters (*e.g.*, $(3, 3, 64)$ in the convolution layer).

To hide key model parameter information, we instead encapsulate parameters into their corresponding generated custom source codes of the obfuscated layer. For example, for a simple one-layer feed-forward neural network, the output can be computed by $Y = \theta(W^T X + b)$, where $X$, $W$, and $b$ is input tensor, parameter of the layer, and bias, respectively. $\theta$ is the activation for neural nodes. For *ModelObfuscator* obfuscation, the network layer can be disguised as $Y = g(X)$, where $g$ is an unknown function. We then implement the correct computation (*i.e.*, $g$) in the generated

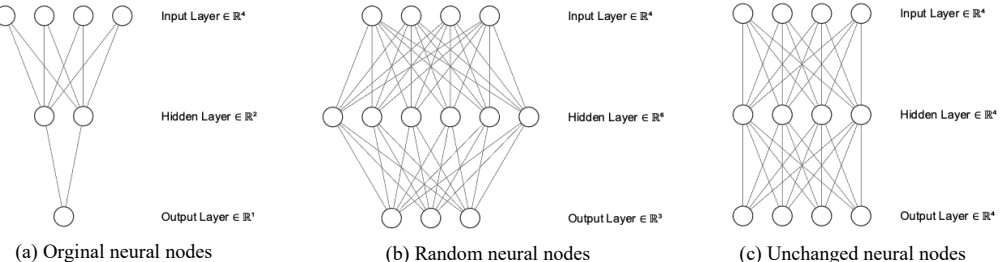

Figure 2: Neural structure obfuscation for a simple feed-forward neural network.

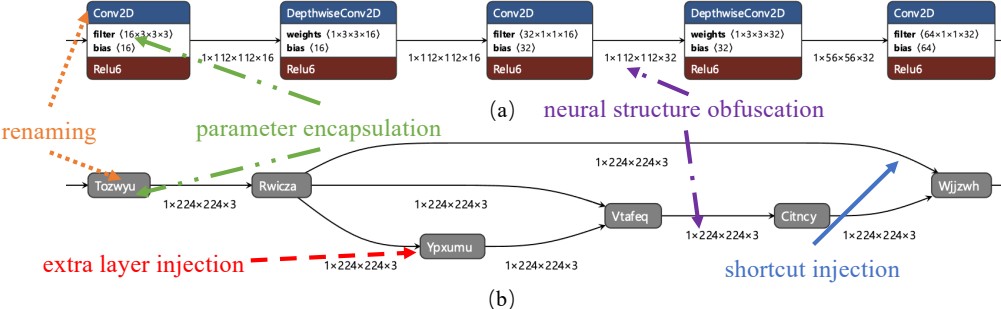

Figure 3: Example of shortcut injection and extra layer injection for the TFLite model extract from a real-world app. (a) part of the original model. (b) the corresponding obfuscated model. This visualization is generated by Netron (Roeder, 2017).

custom TFLite source code, which we then obfuscate. At runtime, function $g$ will be invoked to achieve the computation from $X$ to $Y$. Now an adversary is unable to extract the key parameter information from our obfuscated model. Furthermore, the implementation of $g$ can be obfuscated using transitional and well-proven code obfuscation strategies (Collberg & Thomborson, 2002). After compiling the modified TFLite, adversaries will find it very hard to identify key model parameters by reverse engineering the compiled library.

***Neural structure obfuscation*** Just obfuscating layer names and parameters is not enough, since an adversary may still infer the function of each layer according to the model structure. For instance, Figure 2(a) presents the structure of the neural network, where the input, hidden and output layers include four, two, and one node, respectively. Attacker could search for a surrogate model according to the neural architecture. To solve this problem, *ModelObfuscator* uses *neural structure obfuscation* to obfuscated neural architecture with the goal of confusing the adversary. We propose two strategies for network structure obfuscation: *random* and *align-to-largest*. Given a model with output shapes $s = (s_0, \cdots, s_n)$, where $s_n$ refers to the number of dimensions for the $n$-th channel, the random strategy generate a random shape $\mathbf{r} = (\mathrm{r}_0, \cdots, \mathrm{r}_n)$ of the output for each layer. Figure 2(b) shows an obfuscated model of Figure 2(a) using random strategy. Second, the align-to-largest strategy finds the largest output shape $s'$ and then fill the output shapes of other layers to the size of $s'$. Figure 2(c) shows such an obfuscated model, where the output shapes of each layer are filled up to $(4)$. Note that this will not affect the performance of models because the modified TFLite library will not compute the output using the provided neural structure information.

***Shortcut injection & extra layer injection*** Neural structure obfuscation changes the network structure by inserting new nodes, but the spatial relationships of original layers remain the same. Therefore, even with the above three obfuscation strategies, the adversary can still infer node information by analyzing the spatial relationships of runtime data (*e.g.*, actual input-output values of each node). To further obfuscate the model structure, hence, *ModelObfuscator* applies two more strategies: shortcut and extra layer injection. The injected shortcut and extra layers would destroy the original spatial relationships of TFLite models.

To automatically inject random shortcuts, *ModelObfuscator* first randomly select a shortcut pair $(\mathrm{r}_1, \mathrm{r}_2)$. The outputs of $\mathrm{r}_1$-th layer are then added to the input list of $\mathrm{r}_2$-th layer. For example, the blue solid line in Figure 3(b) is a shortcut inserted into the model extracted from a fruit recognition app, which connects the 2-nd convolution layer and the 5-th convolution layer. Second, to inject extra layers, just like the shortcut injection, *ModelObfuscator* randomly picks a layer pair $(\mathrm{r}'_1, \mathrm{r}'_2)$.

Table 1: The obfuscation error of the proposed model obfuscation method. 'Error': the output difference between the original model and the obfuscated model.

| | ① | ② | ③ | ④ | ⑤ | ⑥ | ⑦ | ⑧ | ⑨ | ⑩ | **Average** |
|---|---|---|---|---|---|---|---|---|---|---|---|
| Error | 0.0 | 0.0 | 0.0 | 0.0 | 0.0 | 0.0 | 0.0 | 0.0 | 0.0 | 0.0 | 0.0 |

The input of the extra layer is the output node list of $r'_1$-th layer, and its output is added to the input list of $r'_2$-th layer. For instance, an extra layer `Ypxumu` (as shown in the red dotted line in Figure 3(b)) is injected between the 2-nd and 3-rd convolution layers.

Note that the injected shortcut and layers will not affect the prediction results of the deployed model because the modified TFLite library will ignore the obfuscation part (*e.g.*, `Ypxumu` layers) in model inference. Specifically, the extra layer $Y = f(X)$ just needs to create the output with a specific shape to confuse the adversary. In addition, extra layer injection will not significantly increase the latency of on-device models because the extra layer does not need much computational resources.

***Combine five obfuscation strategies*** The proposed five obfuscation strategies are applied to the target model sequentially. Figure 3 show an example, where Figure 3(a) is the original model, while Figure 3(b) shows the final obfuscated model. The structure and parameters of the obfuscated model are quite different from that of the original model, which will make the obfuscated model hard to attack. To prevent an adversary from parsing the obfuscated model through reverse engineering the modified TFLite library and customized layers, we use code obfuscation strategies, a well-developed technology to make the code unreadable and unable to be reverse engineered (Appendix B). The combination of model and code obfuscation would hide the key information of models.

### 3.3 MODEL ASSEMBLING AND LIBRARY RECOMPILATION

After obtaining the obfuscated model and modified TFLite source codes, *ModelObfuscator* assembles the new obfuscated model using the obfuscated model structure and then recompiles the modified TFLite library to support the newly generated obfuscated model.

## 4 EVALUATION

### 4.1 EXPERIMENTAL SETTING

**Dataset** To evaluate *ModelObfuscator*'s performance on models with various structures for multiple tasks, we collected 10 TFLite models including a fruit recognition model, a skin cancer diagnosis model, MobileNet (Howard et al., 2017), MNASNet (Tan et al., 2019), SqueezeNet (Iandola et al., 2016), EfficientNet (Tan & Le, 2019), MiDaS (Ranftl et al., 2020), Inception-ResNetV2, PoseNet (Kendall et al., 2015), and SSD (Liu et al., 2016), which are referred to as model ① - ⑩, respectively. The fruit recognition and skin cancer diagnosis model were collected from Android apps (see the provided code repository). The other models were collected from the TensorFlow Hub [3].

**Experimental Environment** *ModelObfuscator* is evaluated on a workstation with Intel(R) Xeon(R) W-2175 2.50GHz CPU, 32GB RAM, and Ubuntu 20.04.1 operating system.

### 4.2 EFFECTIVENESS OF *ModelObfuscator*

We first evaluate the effectiveness of *ModelObfuscator*. Ideally, *ModelObfuscator* should not affect the prediction accuracy of the original models, while providing sufficient defense against model attacks. To this end, we apply all five proposed obfuscation strategies to each model and compare the prediction results based on 1,000 randomly generated inputs. The obfuscation error is calculated as $||y - y'||_2$, where $y$ and $y'$ is the output of original models and obfuscated models, respectively. Note that the number of extra layers and shortcuts is set to 30 in *shortcut injection & extra layer injection*. Table 1 demonstrates that *ModelObfuscator* **model obfuscation strategies have no impact on the prediction results of the original models**.

To show the effectiveness of *ModelObfuscator* in hiding the model's key information, we try to extract the model information of obfuscated models using five software analysis tools. The tools

---
[3] https://tfhub.dev/

Table 2: The success number of existing software analysis tools to extract mobile models with each obfuscation strategy. '*Basic obfuscation*': *renaming + parameter encapsulation*.

|  | TF-ONNX | TFLite2ONNX | TFLite2TF | FlatBuffer | App Attack |
|---|---|---|---|---|---|
| Original | 10 | 9 | 9 | 10 | 8 |
| *Renaming* | 0 | 0 | 0 | 0 | 8 |
| *Parameter encapsulation* | 0 | 0 | 0 | 0 | 2 |
| *Neural structure* | 0 | 0 | 0 | 0 | 8 |
| *Shortcut injection* | 0 | 0 | 0 | 0 | 8 |
| *Extra layer injection* | 0 | 0 | 0 | 0 | 8 |
| *Basic obfuscation* | 0 | 0 | 0 | 0 | 0 |

include TF-ONNX (Developers, 2022) TFLite2ONNX (Wang, 2021), TFLite2TF (Hyodo, 2022), FlatBuffer (Li et al., 2021b), and Smart App Attack (Huang & Chen, 2022), which is proposed to attack the on-device models. In this experiment, we apply the basic obfuscation strategies *renaming* and *parameter encapsulation* on the original models. For model conversion tools, if they can successfully convert the model format, we consider it a successful case for extracting the model information. If *ModelObfuscator* method is effective, these tools cannot work on the models. For the app attacking method, if it can correctly identify the obfuscated model that has the same model structure as the original one on TensorFlow Hub, we consider it a success case. If *ModelObfuscator* is effective, the FlatBuffer extractor cannot parse the information of the obfuscated model and reverse it to the original one. As shown in Table 2, **two basic obfuscation strategies can prevent all existing model extraction tools from parsing the deployed TFLite model**. Besides, parameter encapsulation could prevent App Attack from finding surrogate models on 6 models. When combining renaming and *Parameter encapsulation*, the App Attack on all the models can be prevented. However, except for *parameter encapsulation*, applying other strategies separately does not confuse the App Attack, because App Attack can identify the same model through the parameter comparison.

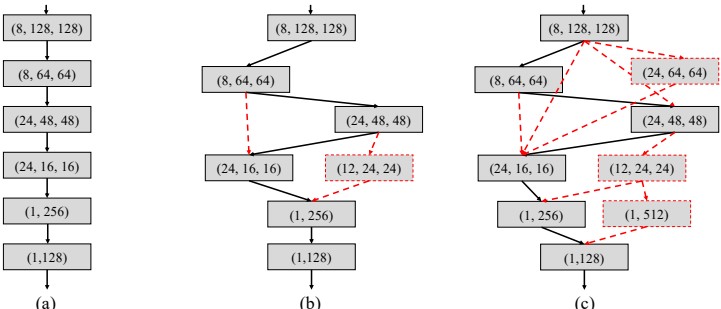

Figure 4: Visualization of structure obfuscation for LeNet. (a) original data flow of LeNet (b) obfuscated model with one shortcut and extra layer (c) obfuscated model with three shortcuts and three extra layers. Red dotted line and red dotted block represent injected shortcut and extra layer.

As we discussed above, adversaries may infer the functionality of each layer by analyzing the data flows of the model. To demonstrate the capability of *ModelObfuscator* in obfuscating the data flows, we show a visualization of LeNet (LeCun et al., 1998) before and after applying *Shortcut injection & extra layer injection*. The visualization of model structure and data flows before and after obfuscation are shown in Figure 4. As can be seen, **it is difficult to recognize the shortcut or extra layer without prior knowledge of the model structure**. When a large number (*e.g.*, more than 30) of shortcuts and extra layers is used to obfuscate the TFLite, the structure of the obfuscated model will become extremely confused (Figure 7). Therefore, the *shortcut* and *extra layer injection* are effective in making it extremely difficult to understand the model structure.

## 4.3 EFFICIENCY OF *ModelObfuscator*

We evaluate the efficiency of *ModelObfuscator* obfuscation strategies by demonstrating the runtime overhead of each model in our dataset. Specifically, we report both the time overhead and memory overhead of the proposed obfuscation method under various settings based on 1,000 randomly generated instances. The results of time and memory overhead are shown in Table 3 and Table 4, respectively. '$(n_1, n_2)$' in the table indicates the the number of shortcuts ($n_1$) and num-

Table 3: Time overhead (seconds per 1000 inputs) of the original model and obfuscated model. We use five obfuscation strategies. '$(n_1, n_2)$': obfuscated models with $n_1$ shortcuts and $n_2$ extra layers.

| | ① | ② | ③ | ④ | ⑤ | ⑥ | ⑦ | ⑧ | ⑨ | ⑩ | **Average** |
|---|---|---|---|---|---|---|---|---|---|---|---|
| Original | 30.3 | 98.5 | 62.1 | 72.5 | 33.4 | 81.2 | 346.6 | 247.4 | 130.4 | 231.2 | **133.4** |
| (0, 0) | 30.4 | 98.7 | 61.1 | 70.4 | 33.5 | 82.3 | 346.4 | 247.0 | 130.7 | 232.1 | **133.3** |
| (30, 0) | 30.7 | 97.9 | 62.3 | 72.7 | 33.1 | 82.0 | 346.9 | 247.1 | 130.2 | 231.0 | **133.4** |
| (0, 10) | 30.2 | 98.6 | 61.1 | 74.1 | 34.9 | 82.3 | 346.7 | 251.5 | 130.9 | 231.6 | **134.2** |
| (0, 20) | 30.6 | 98.3 | 63.3 | 74.1 | 36.8 | 81.3 | 348.2 | 247.1 | 130.5 | 231.3 | **134.2** |
| (0, 30) | 30.4 | 98.5 | 63.2 | 74.6 | 34.5 | 82.8 | 349.0 | 251.9 | 130.4 | 236.7 | **135.2** |

Table 4: Overhead of the model obfuscation on random access memory (RAM) cost (Mb per model). We use five obfuscation strategies. To eliminate the influence of other processes on the test machine, we show the increment of RAM usage for the model inference.

| | ① | ② | ③ | ④ | ⑤ | ⑥ | ⑦ | ⑧ | ⑨ | ⑩ | **Average** |
|---|---|---|---|---|---|---|---|---|---|---|---|
| Original | 18.8 | 49.5 | 33.1 | 46.5 | 41.3 | 51.9 | 241.1 | 373.8 | 39.4 | 97.8 | **99.3** |
| (0, 0) | 18.9 | 53.3 | 33.1 | 49.9 | 45.5 | 56.8 | 244.8 | 381.2 | 40.6 | 99.8 | **102.4** |
| (30, 0) | 18.7 | 51.4 | 32.6 | 49.5 | 46.0 | 55.8 | 244.6 | 381.6 | 40.3 | 100.2 | **102.0** |
| (0, 10) | 25.9 | 58.8 | 42.5 | 53.6 | 49.2 | 69.5 | 248.1 | 386.7 | 55.0 | 107.5 | **109.7** |
| (0, 20) | 30.0 | 67.3 | 50.4 | 56.7 | 56.0 | 66.2 | 253.1 | 387.7 | 66.9 | 112.0 | **114.6** |
| (0, 30) | 35.9 | 81.9 | 55.2 | 57.3 | 71.6 | 73.0 | 267.7 | 397.3 | 73.4 | 123.8 | **123.71** |

ber of extra layers ($n_2$) applied, where $(0, 0)$ indicates only the basic obfuscations (*i.e.*, renaming, parameter encapsulation) and neural structure are involved. We also include the time and memory consumption of the original models as the baseline. As shown in Table 3, even though extra layers are injected into the obfuscated model, ***ModelObfuscator* obfuscated models incur a negligible time overhead** (*i.e.*, approximately 1% on average for the most time-consuming obfuscation). The differences between using various obfuscation settings are also not significant. Because the *parameter encapsulation* will remove some data processing steps in the source code of APIs, the basic obfuscation ('(0,0)' in Table 3) may reduce the latency of TFLite models.

The memory overhead for *ModelObfuscator* obfuscated models is shown in Table 4. To eliminate the impact of different memory optimization methods, we use peak RAM usage where the model preserves all intermediate tensors. It can be seen that the other obfuscation strategies do not affect memory usage except for the *extra layer injection*. Memory usage increases when the number of extra layers increases. Therefore, a trade-off between the complexity of the obfuscation and the memory overhead is worth considering when choosing the obfuscation strategy. Nevertheless, we argue that even with the most complex settings in our experiment, which provide sufficient protection to the original model, **the memory overhead is acceptable (approximately 20%)**.

Table 5: Size change (Mb) of the TFLite library and models after the obfuscation. The size of the obfuscated model is reduced to a few Kb. The original library size is 191.5 Mb. '+' and '-' refer to the increase and decrease, respectively.

| | ① | ② | ③ | ④ | ⑤ | ⑥ | ⑦ | ⑧ | ⑨ | ⑩ |
|---|---|---|---|---|---|---|---|---|---|---|
| library-all | +8.7 | +31.9 | +19.6 | +38.7 | +8.8 | +40.6 | +126.1 | +231.8 | +9.7 | +52.1 |
| model | -5.5 | -16.9 | -10.3 | -17.5 | -5.0 | -18.6 | -66.3 | -121.1 | -5.0 | -27.5 |
| total | +3.2 | +15.0 | +9.3 | +21.2 | +3.8 | +22.0 | +59.8 | +110.7 | +4.7 | +24.4 |
| library-renaming | +8.7 | +31.9 | +19.6 | +38.7 | +8.8 | +40.6 | +126.1 | +231.7 | +9.7 | +52.1 |

Considering that the models are deployed on mobile devices that have limited storage space, we also present the size differences of the modified TFLite software library and the obfuscated models. Note that the size change is caused by creating additional `.so` files to support the inference of the obfuscated layer. Hence, the size difference will be the same with different implementations (*e.g.*, Python, Java). Table5 shows the size change to the TFLite software library and the TFLite models after applying *ModelObfuscator* obfuscation strategies. We use all obfuscation strategies in this experiment and the number of injecting shortcuts and extra layers is 30. Our results show that **the library size change is mainly caused by the *renaming* method**, because it will create a new API for the renamed layer in TFLite library. In addition, the *extra layer injection* will also increase the size of library because it will create the new API to support the extra layer. As the extra layer just

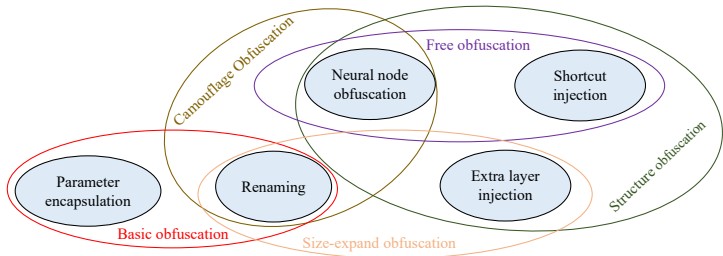

Figure 5: Taxonomy of model obfuscation methods.

has simple function, the affect of the *extra layer injection* is negligible. Our obfuscation strategies **significantly reduce the size of the TFLite model file** because it only keeps the obfuscated minimal structure information. However, **the size of the TFLite library is significantly increased**, and increases the size of the application deployed on the mobile device.

## 5  DISCUSSION

**Taxonomy of model obfuscation methods**  In this paper, we proposed five different model obfuscation strategies. Figure 5 shows a preliminary taxonomy of the different model obfuscation methods and the best practice for model deployment. First, developers can use the *renaming* and *parameter encapsulation* to prevent most model parsing or reverse engineering tools from extracting the information of the deployed model. In the scenarios that computational costs are critical, developers can use the *neural structure obfuscation* and *shortcut injection*, as they do not introduce additional overhead. For structure obfuscation, developers can use *neural structure obfuscation*, *shortcut injection*, and *extra layer injection*. These three methods can significantly increase the difficulty of understanding the data flow of the deployed models. In addition, developers can use *renaming* and *neural structure obfuscation* to disguise the deployed model, which can mislead the adversary into choosing the wrong architecture to produce a surrogate model. If the size of the app package is critical (e.g., deploying the model on devices with limited storage), developers need to carefully consider the trade-off between the number of obfuscated layers leveraging *extra layer injection* with *renaming*. The reason is that if *renaming* is used to create a different obfuscated layer for every layer, *ModelObfuscator* will need to create the corresponding APIs in the TFLite library to support the obfuscated layers, hence increasing the size of the library.

**Limitations**  Although the proposed model obfuscation does not introduce significant computational overhead, it will increase the size of the modified TFLite library. This is because we need to provide support for the new obfuscations made. For a huge network like a 1000-layer network deployed model (although it is unlikely to find such deployed model in real world), the size of the modified TFLite library will significantly increase if we rename every layer. As a result, the app package also increases as the modified TFLite library will also need to be deployed on the device.

**Extracting obfuscated model**  When ModelObfuscator obfuscates the model, it will create a cache file to guide the tool to generate a compatible DL library. Attackers cannot automatically extract the obfuscated model unless they obtain the cache file from the developer's computer (which is unlikely to happen). Generally speaking, attackers must use reverse engineering to get source codes from the compiled library file and cost manual effort to understand them for extracting the obfuscated model.

## 6  CONCLUSION

In this work, we analyzed the risk of deep learning models deployed on mobile devices. Adversaries can extract information from the deployed model to perform white-box attacks and steal its intellectual property. To this end, we proposed a model obfuscation framework to secure the deployment of DL models. We utilized five obfuscation methods to obfuscate the information of the deployed model, *i.e.*, *renaming*, *parameter encapsulation*, *neural node obfuscation*, *shortcut injection*, and *extra layer injection*. We developed a prototype tool *ModelObfuscator* to automatically obfuscate a TFLite model and produce a compatible library with the model. Experiments show that our method is effective in resisting the model parsing tools without performance sacrifice. Considering the negligible extra resources required, and the extra security it achieves, we believe that model obfuscation has the potential to be a fundamental step for on-device model deployment in the future. In future works, optimizing model obfuscation to reduce overhead is worthwhile to be explored.

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

## A    APPENDIX - THE VISUALIZATION OF THE MODEL OBFUSCATION

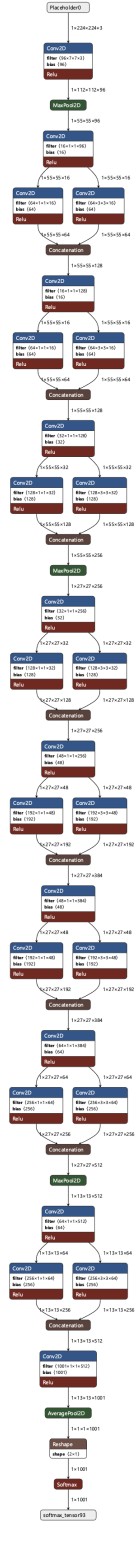

Figure 6: The visualization of the original SqueezeNet model. It is plotted by Netron (Roeder, 2017).

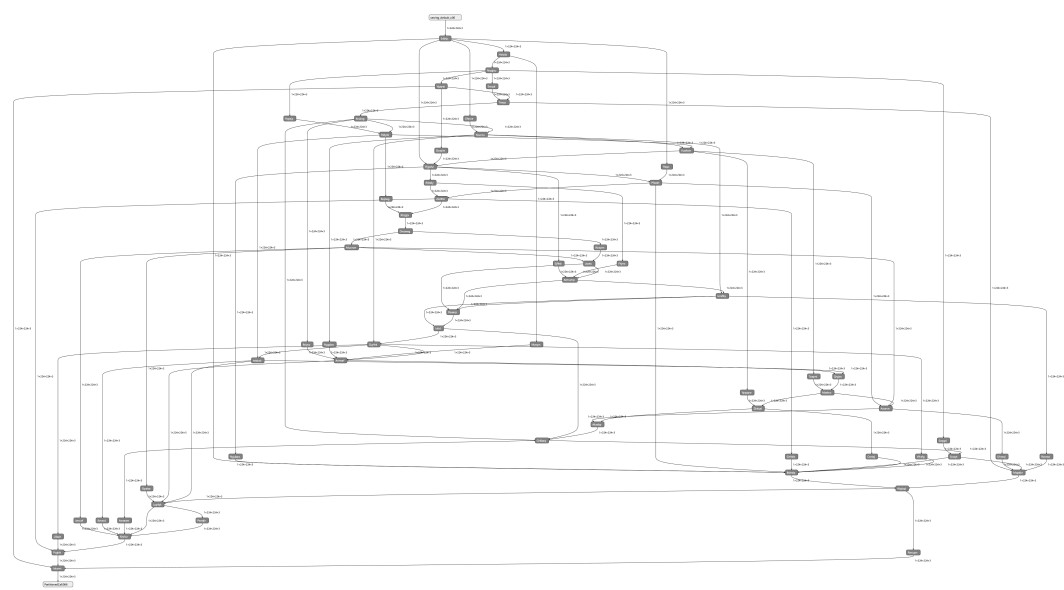

Figure 7: The visualization of the obfuscated SqueezeNet model with all obfuscation strategies. This figure is plotted by Netron (Roeder, 2017). The number of injected shortcuts and extra layers is 30. The obfuscated model can run on the modified TFLite library. The computational time and memory overhead of model obfuscation for this model can be found in Figure 3 and 4. The size change can be found in Figure 5.

## B APPENDIX - EXAMPLE OF CODE OBFUSCATION

The following code is the 'Eval' function of the modified source code to support the `Concatenation` layer. It will finally call the 'EvalImpl' function, which is the detailed implementation for the forward inference.

```
template <KernelType kernel_type>
TfLiteStatus Eval(TfLiteContext* context, TfLiteNode* node) {
  // auto* params =
  //     reinterpret_cast<TfLiteConcatenationParams*>(node->builtin_data);
  // int axis = params->axis;
  TfLiteTensor* output;
  TF_LITE_ENSURE_OK(context, GetOutputSafe(context, node, 0, &output));
  if (IsConstantOrPersistentTensor(output)) {
    // Output is computed in Prepare.
    return kTfLiteOk;
  }
  if (axis < 0) axis += output->dims->size;

  return EvalImpl<kernel_type>(context, node, axis, output);
}
```

After obfuscating the simple code by some simple obfuscation method (*e.g.*, comments removal, functions and variables renaming, whitespaces removal. We use an open-source tool to implement it [4]), the code will become the following form:

```
template<lpurqacghdzw uowpojpkaqnn>TfLiteStatus ojmfjdqxlkrj(TfLiteContext
*zsflmtabfslq,TfLiteNode*node){gqgyxfwmpoax*output;TF_LITE_ENSURE_OK(
zsflmtabfslq,GetOutputSafe(zsflmtabfslq,node,0,&output));if(
IsConstantOrPersistentTensor(output)){return    tsciszklzszy;}if(
dwzzpibyhyhk<0)dwzzpibyhyhk+=output->dims->size;return
tizpeextacqk<uowpojpkaqnn>(zsflmtabfslq,node,dwzzpibyhyhk,output);}
```

---

[4]https://github.com/whoward3/C-Code-Obfuscator

Note that this is just a simple example. For the case when the original model is a 30-layer neural network, it will create more than 10 thousand lines of code for supporting the obfuscated model, and then add the code to the TFLite project to compile a modified TFLite library. If adversaries use the reverse engineering method to extract the source code of the obfuscated layer, ideally they can get the above obfuscated code. But the code will be extremely difficult to read. This is the reason why we think developers can use code obfuscation to resist further reverse engineering for the source code of the modified TFLite library.

