# OpenReview forum: "Model Obfuscation for Securing Deployed Neural Networks"
_ICLR.cc/2023/Conference — Submitted to ICLR 2023_

### Official Review · Reviewer_exnF · 2022-10-15

**Confidence:** 4
**Correctness:** 3
**Technical Novelty And Significance:** 3
**Empirical Novelty And Significance:** 2
**Recommendation:** 5

**Clarity, Quality, Novelty And Reproducibility:**

## Clarity
This work is clearly written and can be easily understood by the reader.

## Quality
The quality of the paper is good.

## Novelty
This work holds some novelty since the topic of model obfuscation is not studied extensively in the community.

## Reproducibility
The authors provided their source code and relevant information in the paper. To the best of my judgment, this work will be reproducible.


**Strength And Weaknesses:**

## Strengths
(+) The topic of deep neural network obfuscation has not been studied extensively in the literature. This work holds novelty in the proposed model obfuscation techniques.
(+) ModelObfuscator achieves convincing results, supporting the claims made by the paper.
(+) The overall presentation of the paper is well-structured and easy to follow.

## Weaknesses
(-) While the authors claim that their proactive defense could disallow a transformation from a non-differentiable model to a differentiable one, in a realistic attack scenario an attacker could simply train a differentiable substitute model by querying the deployed model to create his own dataset. It seems ModelObfusctator cannot prevent this simple attack. It would also be beneficial if the authors could elaborate on their claim why ModelObfuscator could disallow a transformation from a non-differentiable model to a differentiable one, and in which scenarios it is possible and in which it is not.
(-) In essence, the shortcut injection and extra layer injection are decoys to confuse the attacker. However, under the hood, the model architecture is not changed since the library will ignore these parts. It would be more interesting if such an obfuscation could be achieved by really changing the model architecture with the proposed methods.
(-) The authors did not include related work on model obfuscation [A-F]. Consequently, the authors failed to differentiate themselves from the existing literature and to compare their approach against existing methods.
(-) Given, that this work introduces obfuscation tricks, which are not really changing the underlying model architecture to make it more secure, I am not sure if this work is in the scope of the topics for ICLR. This work might be more suitable for a conference on the topic of security & privacy or software engineering.

[A] Anonymizing Machine Learning Models; ESORICS 2021
[B] DeepObfuscation: Securing the Structure of Convolutional Neural Networks via Knowledge Distillation; ArXiv 2018
[C] Mimosanet: An unrobust neural network preventing model stealing; CVPR-W 2019
[D] Preventing Neural Network Weight Stealing via Network Obfuscation; Intelligent Computing. SAI 2020
[E] Hardware-Assisted Intellectual Property Protection of Deep Learning Models; ACM/IEEE Design Automation Conference (DAC), 2020
[F] Preventing DNN Model IP Theft via Hardware Obfuscation; EEE Journal on Emerging and Selected Topics in Circuits and Systems 2021


**Summary Of The Paper:**

This work introduces ModelObfuscator, an obfuscation strategy for deep neural networks. ModelObfuscator introduces 5 model obfuscation strategies, namely renaming, parameter encapsulation, neural structure obfuscation, shortcut injection, and extra layer injection. The authors demonstrate that ModelObfuscator (a) prevents model extraction tools from parsing the deployed model, (b) obfuscated models incur a negligible time overhead with (c) a model overhead of around 20%.

**Summary Of The Review:**

I believe that this work holds some novelty. However, given the insufficient related work and my concern if this work will be of interest to the ICLR community I am currently leaning toward rejection.

---

> ### Author Response · Authors · 2022-11-13
> **Response for Reviewer exnF**
>
> We would like to thank the reviewer for reviewing our paper as well as for the insightful comments. In the following, we will address the questions and comments accordingly:
>
> 1. “an attacker could simply train a differentiable substitute model by querying the deployed model to create his own dataset.”
>
> Yes, that’s true. However, if attackers use queries to reconstruct the target model without any knowledge about the target model, it is actually a black-box attack method. Our proposed method can resist the attackers to extract the original model (without any differences) from the mobile devices to generate white-box attacks. Prior studies like [1] have shown that white-box attacks are much more effective than black-box attacks This will allow many harmful threats to attack mobile devices. Disabling the white-box attacks for the deployed models is also an important improvement for AI safety.
>
>
>       [1] Investigating Top-k White-Box and Transferable Black-box Attack, CVPR2022.
>
> 2. “why ModelObfuscator could disallow a transformation from a non-differentiable model to a differentiable one”
>
> Thanks for your comments. If the attacker wants to transfer a non-differentiable model to a differentiable one, they should know the exact model architecture and weights, then they can assemble such information into a differentiable model format. The proposed model will randomly obfuscate the model architecture and layers of information, and hide the weights in the DL library. Therefore, the only way to achieve such transformation is to reverse the DL library to the source codes and understand the computational details to extract the model architecture and weights (this is almost impossible). This will significantly increase the budgets of attackers. We will clarify this in the revised version.
>
> 3. “the model architecture is not changed since the library will ignore these parts.”
>
> Thanks for your comment. The model architecture (in the model file) is actually changed by the proposed obfuscation. If you mean the attacker can notice the extra architecture due to the empty implementation, it is very hard. This is because the details of each layer are encoded in the implementation of the DL library, attackers must analyze the implementation detail of the DL library to identify the extra layer. Adding extra layer is in analogy with adding dead code in code obfuscation, which may confuse attackers. Therefore, we believe architecture obfuscation is a promising way to secure the model. We cannot add too much engineering detail to the paper, but we will try to clarify it in the revised version.
>
>
> 4. “Missing some related works”
>
> For [A, B, C, D], they propose defences on different levels (i.e., algorithm level and side-channel level). However, these works cannot resist the attackers using software analysis methods to extract the original model from the device and disable the attacker generates white-box attacks. For [E,F] these methods cannot be applied to all edge devices such as some Andoird smartphones, they need additional hardware to support the protection. The proposed method can be applied to almost all mobile devices without any hardware-specific steps for developers. We will add them to the related work discussion.
>
>
> 5. “ if this work is in the scope of the topics for ICLR.”
>
> Currently deploying the model on devices gains popularity in software development. It is very dangerous that attackers can directly get the original model without any losses. We believe how to make the deployed model has similar safety to the remote model should be fundamental research for AI deployment. We think ICLR is a good venue to share our idea with AI researchers and developers to improve the safety of deployed AI models.

---

### Official Review · Reviewer_17Yg · 2022-10-20

**Confidence:** 4
**Correctness:** 4
**Technical Novelty And Significance:** 3
**Empirical Novelty And Significance:** 3
**Recommendation:** 8

**Clarity, Quality, Novelty And Reproducibility:**

The paper is well-written and details about obfuscation strategies are easy to understand.
The content of the paper is novel, and reproducibility is assured thanks to the inclusion of the code (anonymous link inside the paper)

**Strength And Weaknesses:**

**Strenghts**

+ novel manipulations of neural networks that preserve the original performance, but rearranging and complicating the structure
+ the core contribution of the paper can help bringing the gap between software obfuscation techniques and AI technologies
+ converters can not deal with this new format of models
+ open-source code is released (but maybe the authors could have submitted a Dockerfile that would have make the testing easier)

**Weaknesses**

+ the authors do not show how their manipulations are more effective than regular Android obfuscation techniques. In particular, the code-related manipulations (such as the class renaming) are not novel. An example of previous implementations can be found in ObfuskAPK [a]. The authors might compare their techniques, showing that regular tools can not obfuscate part of the machine learning model.
+ converters do not work because this obfuscation is not known yet. Hence the authors should add this limitations to their work, or at least discuss what can be done once this obfuscation technique is known and how converters could be patched to take into account this technique.

[a] Aonzo, S., Georgiu, G. C., Verderame, L., & Merlo, A. (2020). Obfuscapk: An open-source black-box obfuscation tool for Android apps. SoftwareX, 11, 100403.

**Summary Of The Paper:**

The authors propose ModelObfuscator, a methodology that applies novel obfuscation techniques customised to hide relevant information regarding architecture and operations of the input neural network.
Among the obfuscation, ModelObfuscator can hide the real shape of the layers of the input neural network, re-arrange the data flow inside layers (but preserving the original one), and rename classes and functions.
The effectiveness and preciseness of the original model is preserved, with a negligible overhead.

**Summary Of The Review:**

+ Obfuscation techniques customised for hiding meaningful information of deployed model, such as architecture and layers' shape
+ Original performance is assured with negligible overhead

---

> ### Author Response · Authors · 2022-11-13
> **Response for Reviewer 17Yg**
>
> We would like to thank the reviewer for reviewing our paper as well as for the insightful comments. In the following, we will address the questions and comments accordingly:
>
> 1. “Compare the existing code obfuscation and the proposed model obfuscation.”
>
> Thanks for your comments. Our proposed obfuscation method has five obfuscation strategies. The renaming strategy is similar to the renaming operation in code obfuscation. However, the other obfuscation strategies (i.e., parameter encapsulation, neural structure obfuscation, shortcut injection, and extra layer injection) are model-specific obfuscation strategies and are different from the obfuscation operations in code-related tasks. If we just use the code-related obfuscation method, we cannot obfuscate the spatial relation of the layers and hide the parameters in the model. As the model design usually has some common architectures, attackers may easily estimate the structure of the naively obfuscated model and extract the weights from the model. Therefore, we need some model-specific obfuscation strategies to further obfuscated the weights, model architecture, and neural connections. We will add a discussion of such a comparison in the related works.
>
> 2. “authors should add this limitation to their work, or at least discuss what can be done once this obfuscation technique is known and how converters could be patched to take into account this technique.”
>
> Thanks for your comment. Because our obfuscation method randomly obfuscates the model detail as the information which is not understandable but hides the computational detail in the compiled binary files (DL APIs library). If attackers want to reverse the obfuscated model to the original model, they should identify the connection between the obfuscated layers and the original layers. However, such connections are totally random and generated by the obfuscator on the developer’s side. The only thing attackers can do is use reverse engineering to get source codes from the compiled library file and cost manual effort to understand the code (the code also can be obfuscated). However, when the ModelObfuscator obfuscates the models, it will create a cache file (it can be deleted after the obfuscation) to guide the tool to generate a compatible DL library. If the attackers get the file from the developer's machine, they can follow the rules to reconstruct the original model. We will clarify this in the discussion.
>
> 3. “maybe the authors could have submitted a Dockerfile that would make the testing easier”
>
> Thanks for your comments. We will release the Dockerfile in our GitHub repository.

---

### Official Review · Reviewer_pBz1 · 2022-10-24

**Confidence:** 4
**Correctness:** 1
**Technical Novelty And Significance:** 1
**Empirical Novelty And Significance:** 2
**Recommendation:** 3

**Clarity, Quality, Novelty And Reproducibility:**

**Novelty**
- The obfuscation strategies proposed in the paper seem trivial.
- [Xu et al., 2018] seems closely related to the work in this paper, however it is not cited by the paper. How does ModelObfuscator compare to this prior work?

**Quality**

*Quality - related works:*
- Cited references on adversarial machine learning seem outdated, presenting defenses that have been proven ineffective as being state-of-the-art (e.g., adversarial example detection, cited as [Ma et al., 2018] in the paper).
- The related works section seems to only cite methods for producing adversarial examples in terms of attacks. This seems incomplete and skewed, considering that the paper is trying to encompass all types of attacks on devices.

*Security of machine learning:*
- The paper loosely tackles the notion of attacks against machine learning models on edge devices and claims that obfuscation is the solution to all of them. Instead, a threat model should be explicited, clearly stating what attacks are a risk on edge devices, which can be defended with obfuscation and it can be done. Moreover, the assumptions about the attacker and defender capacities and objectives should be stated.
- Multiple technical inaccuracies are present in the paper:
  * If the attacker has access to the model and forward pass, it is unclear how the defender can prevent differentiation. Most gradient obfuscation techniques can be defeated by gradient approximations. Moreover, none of the obfuscation techniques introduced in the paper seem to alter the gradients.
  * It is stated that model inversion is a white-box attack (Sec. 3.2). However, it should be clear that if the attacker has access to the model (which is what white-box means), they do not need model inversion anymore; they have the actual model. Model inversion only applies to black-box access, e.g., machine learning as a service (MLaaS) and prediction APIs.
  * Even if the attacker did not have white-box access to the model, model inversion most commonly has a goal to reproduce the functionality of a model in a surrogate. It is known that the actual architecture or hyperparameters do not matter, and the model can be leaked anyway with high fidelity (e.g., [Orekondy et al., 2020]). Based on the surrogate model, most other types of attacks can be performed (model inversion for intellectuall property loss, followed by adversarial examples and membership inference attacks). The paper does not address this aspect.
- Due to the lack of an explicit threat model, it it unclear what happens if the attacker is aware of the obfuscation strategy that is being used.
- The terminology used in the paper around machine learning security is not standard and not meaningful, e.g., trustworthy training, proactive defenses.

*Neural network obfuscation as security measure* Based on the points above, it is unclear how model obfuscation helps with any of the attacks used as motivation for the paper. It seems that the attacker has white-box access to the model on the device, which, obfuscated or not, is able to provide correct predictions. The attacker can use that model directly or build a simpler, non-obfuscated surrogate reproducing its functionality, that can be used for the attacker's end goal.

*Necessity for neural network obfuscation* Beyond the application of model obfuscation for security goals, I challenge the need for obfuscation in neural networks in a broader sense. As recognized in the paper, obfuscation started out applied to code and programming. There, the notion makes sense, as code is (mostly) written by humans and made to be understood by humans. Obfuscation makes code unreadable to humans. Or, neural network architectures are not particularly intuitive, and often enough the result of an automated optimization process. As such, the task of obfuscating them might not have many applications today.

*Quality of experiments:*
- Almost none of the experiments performed in the paper are related to the goal stated for the proposed obfuscation, which is the security of deployed models.
- A valid evaluation would be to craft attacks from each category that ModelObfuscator is trying to defend (e.g., adversarial example, membership inference).
- SmartAppAttack should be cited and briefly explained earlier in the paper.
- The paper does not provide a deeper analysis as to why model parsers fail on the obfuscated models. It might be for trivial reasons, such as the fact that these very recent parsers (2021-2022) are in preliminary stage of development, do not support custom layers, etc. It is also unclear what this experiment proves, as, for all practical purposes, the obfuscated models seem to still be functional.

**Clarity**
- The writing style needs improvements, as some sentences do not make sense.

**Reproducibility**
- The implementation of ModelObfuscator is open-source.

**References**
* [Xu et al., 2018] Hui Xu, Yuxin Su, Zirui Zhao, Yangfan Zhou, Michael R. Lyu, Irwin King. DeepObfuscation: Securing the Structure of Convolutional Neural Networks via Knowledge Distillation. 2018.
* [Orekondy et al., 2020] Tribhuvanesh Orekondy, Bernt Schiele, Mario Fritz. Prediction Poisoning: Towards Defenses Against DNN Model Stealing Attacks. ICLR 2020.



**Strength And Weaknesses:**

Strengths:
- The paper proposes a method and implementation for obfuscating TFLite models.
- The experiments show that models that use ModelObfuscator cannot be parsed by model parsers.
- The implementation of ModelObfuscator is open-source.

Weaknesses:
- The motivation of the paper related to model security is loosely covered, inexact, and not followed throughout the paper.
- The experiments do not evaluate the proposed obfuscation w.r.t. the main motivation of the paper, security goals.
- More generally, the paper fails to motivate the need for neural network obfuscation, as neural network architectures are not particularly interpretable and sometimes even generated automatically via neural architecture search.
- The paper seems to lack scientific novelty and can be viewed, at best, as an engineering feat (trivial obfuscation strategies).

Please see details below.

**Summary Of The Paper:**

Motivated by the security of machine learning models running on device, this paper proposes ModelObfuscator, an obfuscation technique designed to protect models from multiple types of attacks, including adversarial examples, model extraction and membership inference. The proposed obfuscation includes five types of model changes, ranging from layer renaming, introduction of new layers and connections, and operation size masking. The correct predictions of the neural network are maintained after the obfuscation process. ModelObfuscator is implemented for TFLite models. Experiments are performed on ten TFLite architectures.

**Summary Of The Review:**

Paper of limited scientific contribution, with weak motivation and poor evaluation.

---

> ### Author Response · Authors · 2022-11-13
> **Response for Reviewer pBz1**
>
> 12. “A valid evaluation would be to craft attacks from each category that ModelObfuscator is trying to defend (e.g., adversarial example, membership inference).”
>
> Thanks for your comment. However, our goal is to disable the model extraction for the deployed models. The proposed method can successfully obfuscate the model (Table 2), which means attackers only can generate black-box attacks for it. Although our method can reduce the attack performance (from white-box setting to black-box setting),  such comparison (black-box vs white-box) is not in the scope of this study.
>
> 13. “SmartAppAttack should be cited and briefly explained earlier in the paper.”
>
> Thanks for your comment. We cited and discussed that paper in the Introduction section and Experiment section (Page 1 and 7)
>
> “The paper does not provide a deeper analysis as to why model parsers fail on the obfuscated models.”
> The reason why the model parser tools fail on the obfuscated models is the details (layers name, architectures) of obfuscated models are randomly generated by the proposed method. It is not expectable and understandable. So the software analysis tool cannot work on the obfuscated models

---

> > ### Comment · Reviewer_pBz1 · 2022-11-22
> > **Thank you for your response**
> >
> > I would like to thank the authors for their detailed response. I will answer some of the points in the following.
> >
> > 2 & 11. “Almost none of the experiments performed in the paper are related to the goal stated for the proposed obfuscation.”
> >
> > > The evaluation of the proposed method for resisting the model extraction attacks using software analysis can be found in Table 2. Also see the answer to question 2 for more details.
> >
> > None of those seem to be model extraction attacks. The only baseline designed towards security goals is SmartAppAttack, which seems steered at adversarial examples. An attacker could most likely defeat the proposed obfuscations quite easily if they tried. It is the due diligence of the paper to prove that ModelObfuscator can reach its goal and withstand (at least) a minimal intentional attack.
> >
> > 8. “Model inversion only applies to black-box access”, “ model inversion most commonly has a goal to reproduce the functionality of a model in a surrogate.”
> >
> > > Thanks for your point. However, maybe you are referring to the model stealing and model extraction attacks. Model inversion attacks’ goal is to obtain the sensitive information of the training samples [6,7].
> >
> > You are right, point taken.
> >
> > 9. “The attacker can use that model directly or build a simpler, non-obfuscated surrogate reproducing its functionality”
> >
> > > Yes, that’s true. Attackers could rebuild a surrogate model by quering the target model. However, such attack is actually black-box attack, which cannot reconstruct the model that is the exactly same as the original one. Prior studies like [8] have shown that white-box attacks are much more effective than black-box attacks. Our proposed method is mainly designed for defend write-box attacks, i.e., detering attackers from directely obtaining original model from mobile devices. Disabling the white-box attacks for the deployed models is also an important improvement for AI safety. We will clarify this in the revised version.
> >
> > >  [8] Investigating Top-k White-Box and Transferable Black-box Attack, CVPR2022.
> >
> > Here again it seems the answer, like the paper, is mixing different types of attacks. Two types of attacks are not the same just because they are both performed with the same level of access (e.g., white-box or black-box). My statement refers to model theft (producing a model surrogate of same functionality). This only makes sense in a black-box scenario, in white-box there is nothing to steal. Ref [8] you are proposing as counterexample refers to adversarial examples. Please refer back to the references I included in the review and citations there-in, where it is shown that model theft is successful in leaking model functionality with high fidelity without knowing the architecture of the stolen model. As such, the surrogate produced by model theft is the first step for producing white-box adversarial examples, stealing intellectual property, and other attacks, that ModelObfuscator would not be able to prevent. These two-step attacks are standard community practice and challenge the answer to point 12.:
> >
> > > ...However, our goal is to disable the model extraction for the deployed models.
> >
> > This goal does not seem to have been accomplished.
> >
> > While I appreciate the authors' answer, my questions and comments have not been addressed. The discussion seems a bit circular, each party reiterating their initial points with little progress.

---

> > > ### Author Response · Authors · 2022-11-23
> > > **Thanks for your reply**
> > >
> > > Thanks for your reply. However, there are still some points we want to clarify:
> > >
> > > 1. “None of those seem to be model extraction attacks.”
> > >
> > > Currently, extracting the model's key information from software stacks is straightforward, which does not require complex extraction attack strategies. Because the TFLite platform uses a simple defense method that disables the back-propagation function, attackers cannot directly generate white-box attacks on deployed models. The widely-used tools like TF-ONNX, and TFLite2TF can convert the TFLite model to other model formats, which can be considered as a model extraction method at the software level for generating white-box attacks.  In addition, existing methods like FlatBuffer-level reverse engineering and Smart App Attack achieve a high success rate in information extraction. Our evaluation (in the updated Table 2) shows our obfuscation can defend such model extractions. As far as we know, there are no other extraction attack strategies that can obtain the model's key information by reverse-engineering the software stacks.
> > >
> > >
> > > 2. “ the surrogate produced by model theft is the first step for producing white-box adversarial examples, stealing intellectual property, and other attacks, that ModelObfuscator would not be able to prevent. ”
> > >
> > > The two-step attack you mentioned can be found in [1] and it was classified as a black-box attack (i.e., does not use the inner information of deployed models). Generating surrogate models is actually a black-box process, which cannot guarantee the surrogate model is the same as the original one, hence achieving worse attack results than white-box attacks (extracting the deployed model using software analysis). In addition, it is true that our method cannot prevent all kinds of attacks. Our method can effectively defend against the attacks specified in the threat model. We added the discussion of the defence approach you mentioned in related work.
> > >
> > >       [1] Papernot, Nicolas, et al. "Practical black-box attacks against machine learning." Proceedings of the 2017 ACM on Asia conference on computer and communications security. 2017.

---

> ### Author Response · Authors · 2022-11-13
> **Response for Reviewer pBz1**
>
> 4. “The obfuscation strategies proposed in the paper seem trivial.”
>
> This is not the case - several non-trivial obfuscations are proposed. We analyse the pattern of mainstream DL frameworks, the model information is saved in the model file and it will call the DL APIs to run the model. Then, we propose an end2end obfuscation strategy for current mainstream DL platforms like TensorFlow Lite. The ModelObfuscator will randomly obfuscate the model, and automatically generate a corresponding DL library which supports the model inference. In addition, we propose five obfuscation methods. Overall, we first propose an end2end solution for obfuscating the deployed model at software level, which can be used for the current DL platform but also can be adapted to future software development.
>
> 5. “[Xu et al., 2018] seems closely related to the work in this paper, however, it is not cited by the paper. How does ModelObfuscator compare to this prior work?”
>
> Thanks for your comment. This work is relevant but their goal is totally different from us.  Xu et al., 2018 propose a method that simulates the feature extractor of CNN with a shallow and sequential convolutional block. First, this method train a simulation model with different CNN architecture that has similar performance. It can prevent the attacker knows the original CNN design, but it cannot disable the attacker to extract the model from the software. Second, the method only works on the CNN model. In contrast, our method can be used to protect all DL models and disable the software analysis tool to parse the information of the model. We will discuss this work in the related work in the updated version.
>
> 6. “the related work section is not complete”
>
> Thanks for your suggestion. This study refers to knowledge of various areas. Although it is hard to include all SOTA methods. We will update the related work section and include more recent studies in our revised version.
>
> 7. “it is unclear how the defender can prevent differentiation”
>
> Our obfuscation method can prevent differentiation because 1) existing on-device models only support forward propagation, 2) to perform differentiation, it is required to reconstruct the on-device model to a format supporting backward propagation, and 3) ModelObsfucator prevents extracting model’s key information and reconstructing a differentiable model. We will clarify this in our revision.
>
> 8. “Model inversion only applies to black-box access”, “ model inversion most commonly has a goal to reproduce the functionality of a model in a surrogate.”
>
> Thanks for your point. However, maybe you are referring to the model stealing and model extraction attacks. Model inversion attacks’ goal is to obtain the sensitive information of the training samples [6,7].
>
>       [6] Model inversion attacks that exploit confidence information and basic countermeasures. Proceedings of the 22nd ACM SIGSAC conference on computer and communications security.
>
>       [7] The secret revealer: Generative model-inversion attacks against deep neural networks. Proceedings of the IEEE/CVF conference on computer vision and pattern recognition. 2020
>
> 9. “The attacker can use that model directly or build a simpler, non-obfuscated surrogate reproducing its functionality”
>
> Yes, that’s true. Attackers could rebuild a surrogate model by quering the target model. However, such attack is actually black-box attack, which cannot reconstruct the model that is the exactly same as the original one. Prior studies like [8] have shown that white-box attacks are much more effective than black-box attacks. Our proposed method is mainly designed for defend write-box attacks, i.e., detering attackers from directely obtaining original model from mobile devices. Disabling the white-box attacks for the deployed models is also an important improvement for AI safety. We will clarify this in the revised version.
>
>       [8] Investigating Top-k White-Box and Transferable Black-box Attack, CVPR2022.
>
> 10. “the task of obfuscating them might not have many applications today.”
>
> Deploying ML models on mobile devices is gaining popularity recently because it will avoid the transmission of sensitive data between the user’s device and server. However, one major issue of the current deployment strategy is that attackers can extract the original model from the app package and conduct white-box attack. The proposed technique can obfuscate the model file. It can help the deployed model gains safety similar to the remote model. We will further clarify this in the introduction and discussion.
>
> 11. “Almost none of the experiments performed in the paper are related to the goal stated for the proposed obfuscation.”
>
> The evaluation of the proposed method for resisting the model extraction attacks using software analysis can be found in Table 2. Also see the answer to question 2 for more details.

---

> ### Author Response · Authors · 2022-11-13
> **Response for Reviewer pBz1**
>
> We would like to thank the reviewer for reviewing our paper as well as for the insightful comments. In the following, we will address the questions and comments accordingly:
>
> 1. “The motivation of the paper and threat model.”
>
> Threat model: The on-device model is usually saved as a separate file (e.g., .tflite file) and packed into the app package. The attackers can either download the target app from the app markets (e.g., Google Play and iOS App store) or extract the app package file (e.g., APK file for Android, and IPA file for iOS) from hosting devices. These app package files can then be decompiled by off-the-shelf reverse-engineering tools (e.g., Apktool [1] and IDA Pro [2]) to get the original DL model file. Although many on-device DL platforms do not support some advanced functions like backpropagation, attackers can assemble the model architecture and weights into a differentiable model format, or they can use software analysis methods to generate attacks for the target model file [7, 8].
>
> In this paper, we focus on obfuscating the information of the model to disable the software analysis and model conversion tools from extracting model’s information to conduct white box attack. We will clarify  the threat model in the Methodology section.
>
>       [1] https://ibotpeaches.github.io/Apktool/
>
>       [2] https://hex-rays.com/ida-pro/
>
>       [3] Smart app attack: Hacking deep learning models in android apps. IEEE Transactions on Information Forensics and Security, 2022.
>
>       [4]  Deeppayload: Black-box backdoor attack on deep learning models through neural payload injection. In 2021 IEEE/ACM 43rd International Conference on Software Engineering (ICSE), 2021
>
> 2. “The experiments do not evaluate the proposed obfuscation w.r.t. the main motivation of the paper, security goals.”
>
> We evaluated the proposed obfuscation’s ability to increase model security by measuring whether attackers could obtain the key information (like model structure, parameters) of models when they have access to the models. Preventing attackers from obtaining key model information can against a lot of white-box attacks. We will clarify this in the updated version.
>
> Table 2 in the paper shows the performance of the proposed tools for to disabling the software analysis and model conversion tools from extracting model’s information. Specifically, we have 3 kinds of methods to extract the model's key information. First,  attackers can use model conversion to extract the model to their needs, such as TF-ONNX, TFLite2ONNX, and TFLite2TF. Second,  attackers can extract the feature of the model details (i.e., architecture and weights), then find a similar model on the Internet, and fine-tune the collected model (such as the App Attack). Third, the most commonly used on-device model format TFLite runs on the FlatBuffers platform (https://google.github.io/flatbuffers/). Attackers can extract model information on the FlatBuffers level. The following table shows that our technique could disable all the approaches to extract model’s key information with basic obfuscation strategies.
>
> |                          | TF-ONNX | TFLite2ONNX | TFLite2TF | Flatbuffer | App Attack |
> |--------------------------|---------|-------------|-----------|------------|------------|
> | Original                 | 10      | 9           | 9         | 10         | 8          |
> | Renaming                 | 0       | 0           | 0         | 0          | 8          |
> | Parameter  encapsulation | 0       | 0           | 0         | 0          | 2          |
> | Neural  structure        | 0       | 0           | 0         | 0          | 8          |
> | Shortcut injection       | 0       | 0           | 0         | 0          | 8          |
> | Extra layer injection    | 0       | 0           | 0         | 0          | 8          |
> | Basic obfuscation        | 0       | 0           | 0         | 0          | 0          |
>
>       [5] DeepPayload: Black-box Backdoor Attack on Deep Learning Models through Neural Payload Injection. International Conference on Software engineering, 2021.
>
> 3. “More generally, the paper fails to motivate the need for neural network obfuscation, as neural network architectures are not particularly interpretable and sometimes even generated automatically via neural architecture search.”
>
> The non-interpretability of the neural networks you mentioned is on the human level. However, one major issue with most  deployed models is that currently everyone can get the original model file on mobile devices like Android, no matter the interpretability of the model. It is true that their architectures are not interpretable, but attackers can use the extracted models to perform white-box attacks on the ML components of the apps. Our goal is to obfuscate the model in the model file and in the source code. With the help of our techniques, model owners could prevent the obfuscated deployed models from being white-box attacked or stolen by malicious users.

---

### Official Review · Reviewer_Lpda · 2022-10-26

**Confidence:** 4
**Correctness:** 3
**Technical Novelty And Significance:** 2
**Empirical Novelty And Significance:** 2
**Recommendation:** 5

**Clarity, Quality, Novelty And Reproducibility:**

I like this paper as it studies a research problem that could be the problem soon. Reverse-engineering the models deployed to the edge may offer the white-box advantage to the attacker. It could be much easier to make other attacks on the model such as privacy attacks or adversarial attacks. Thus, the paper studies a defense.

However, there are three major problems of the proposed approach:

(1) Obfuscation cannot be a defense against reconstruction attacks. The paper cited several papers and insights from the software-engineering community, but those papers are unlikely to study code obfuscation as a defense. Rather, they study the obfuscation attacker to understand the limits of malware writers who aim to increase the time and effort of security analysts.

As a counter-example, the model extraction attacker who just queries the model to extract the parameters can still extract parameters precisely without reverse-engineering the model from the software stack.

Moreover, (let's assume that we exclude the extraction attacks) if the victim really wants to obfuscate the model parameters, one way is to encrypt them completely by utilizing TEEs. By doing so, the attacker who naively reconstructs the model parameters at the software level cannot get the actual parameters.

Thus, I recommend those changes:

> Clarify the threat model that the reconstruction attacker operates in.
> Tone down the entire paper; ModelObfuscator is not a defense or doesn't work against all reconstruction attacks.
> Clarify the goal of the mitigation mechanism.


(2) There should be an evaluation against reconstruction attacks. Prior work proposes three types of reconstruction attacks: (1) the reconstruction of model parameters via query, (2) the reverse-engineering of the model from the software stack, and (3) the reconstruction of model architecture from side-channel information. This paper claims that ModelObfuscator can be a defense against reconstruction attacks, but there is no evaluation against any of those adversaries.

Thus:

> I'd like to see the results on the effectiveness of ModelObfuscator against those three attacks.


(3) There should be an evaluation against adaptive attacks. A conventional way to test the obfuscation mechanism is to assume that the attacker also has access to the same obfuscator (this is also used to evaluate the security of encryption mechanisms). Access to the obfuscator should not reduce the time and effort of the adversary who wants to reverse-engineer the victim model.

Thus:

> I'd like to see the results of a scenario where the adversary can also use ModelObfuscator.

**Details Of Ethics Concerns:**

No concern.

**Strength And Weaknesses:**

Strengths:

1. The paper studies a research problem that may carry some importance in the future.
2. The paper proposes ModelObfuscator that employs six different ways to obfuscate a model.
3. The paper shows the proposed mechanism does not change a model's behaviors nor increase its computations.


Weaknesses:

1. The approach relies on "security by obscurity."
2. The evaluation against existing reconstruction attacks and adaptive attacks should be done.

**Summary Of The Paper:**

This paper raises a bar against an adversary who aims to reverse-engineer efficient models deployed to edge devices. A naive deployment practice allows the attacker to just extract the model's architecture and parameters which work as a prior for other attacks (such as adversarial attacks). To address this issue, the paper proposes ModelObfuscator that converts a victim model's architecture and parameters. It also obfuscates the ML frameworks by obfuscating the function names (corresponding to layers) and the ways they perform the layerwise computations. In evaluation, the paper shows that the resulting models return the same outputs on a set of input samples even after applying the obfuscation, and the obfuscation method does not incur large extra computations.

**Summary Of The Review:**

This paper studies a research problem that can be important in the future (as we deploy more and more models to edge devices). To address this issue, the paper proposes a method that obfuscates the model internals (such as architectures, parameters, and computation details).

However, this defense has the following major problems:

(1) The defense heavily relies on "security by obscurity."
(2) The defense was not evaluated against any reconstruction attacks.
(3) The defense was not evaluated against an adaptive adversary.

Thus, I believe this paper (as-is) has more flaws than strengths. But, I am happy to update my review score if the authors address all the requested changes.

---

> ### Author Response · Authors · 2022-11-13
> **Response for Reviewer Lpda**
>
> 5. “ the results on the effectiveness of ModelObfuscator against those three attacks.”
>
> Thanks for your comments.  Defense against the reconstruction of model parameters via query and side-channel information is not in the scope of this study.  Our obfuscation and the other defences actually protect the model on different levels. Our study focuses on how to resist model extraction attacks using software analysis.
>
> Although Table 2 shows the performance of the proposed tools for resisting the model extraction using software analysis, we should clarify the setting of the experiment. Currently, we have 3 kinds of methods to extract the model using software analysis. First,  attackers can use model conversion to extract the model to their needs, such as TF-ONNX, TFLite2ONNX, and TFLite2TF. Model conversion methods utilize the APIs provided by the DL library to obtain the model details (layer types and spatial relations), and assemble them in other DL platforms to enable the advanced analysis for on-device models. Second,  attackers can extract the feature of the model details (i.e., architecture and weights), then find a similar model on the Internet, and fine-tune the collected model (such as the App Attack). Third, the most commonly used on-device model format TFLite runs on the FlatBuffers platform (https://google.github.io/flatbuffers/).  Attackers can extract model information on the FlatBuffers level.
> We will add a ‘FlatBuffers’ reverse engineering attack [9] in Table 2 to perform the third kind of model extraction attack using software analysis. A successful ‘FlatBuffers’ attack can parse the information of the obfuscated models on the FlatBuffers level and reverse it to the original model. The updated experiment is shown as follows:
>
> |                          | TF-ONNX | TFLite2ONNX | TFLite2TF | Flatbuffer | App Attack |
> |--------------------------|---------|-------------|-----------|------------|------------|
> | Original                 | 10      | 9           | 9         | 10         | 8          |
> | Renaming                 | 0       | 0           | 0         | 0          | 8          |
> | Parameter  encapsulation | 0       | 0           | 0         | 0          | 2          |
> | Neural  structure        | 0       | 0           | 0         | 0          | 8          |
> | Shortcut injection       | 0       | 0           | 0         | 0          | 8          |
> | Extra layer injection    | 0       | 0           | 0         | 0          | 8          |
> | Basic obfuscation        | 0       | 0           | 0         | 0          | 0          |
>
>       [9] DeepPayload: Black-box Backdoor Attack on Deep Learning Models through Neural Payload Injection. International Conference on Software engineering, 2021.
>
> 6. “The results of a scenario where the adversary can also use ModelObfuscator.”
>
> Thanks for your comments.  Even attackers have the access to ModelObfuscator, they cannot perform such adaptive attacks for ModelObfuscator. This is because our obfuscation method randomly obfuscates the model detail and hides the computational detail in the compiled binary files (DL APIs library). To reverse the obfuscated model to the original model, the connection between the obfuscated layers and the original layers must be identified. However, such connections are totally random and generated by the obfuscator on the developer’s side. It is like the code obfuscation method, not the encryption method. In other words, the obfuscated model does not need a decrypt process in the runtime environment. We will clarify this in our revised version.

---

> ### Author Response · Authors · 2022-11-13
> **Response for Reviewer Lpda**
>
> We would like to thank the reviewer for reviewing our paper as well as for the insightful comments for improvements. In the following, we will address the questions and comments accordingly:
>
> 1. “Obfuscation cannot be a defense against reconstruction attacks”
>
> Yes, that’s true. Obfuscation cannot defend attacks by querying to reconstruct the target model. However, such attack is actually black-box attack which cannot reconstruct the model that is exactly the same as the original one. Prior studies like [1] have shown that white-box attacks are much more effective than black-box attacks. Our proposed method is mainly designed for defending against write-box attacks, i.e., detering attackers from directely obtaining original model from mobile devices. Disabling the white-box attacks for the deployed models is also an important improvement for AI safety. We will clarify this in the revised version.
>
>
>       [1] Investigating Top-k White-Box and Transferable Black-box Attack, CVPR2022.
>
> 2. “The cited code obfuscation papers are unlikely to study code obfuscation as a defense.”
>
> Thanks for your comment. Code obfuscation actually can be used in two scenarios [2]: hide malware’s harmful functionality and against reverse engineering of commercial software [3]. ModelObfuscator retargets the last use scenarios of code obfuscation to protect deployed models. We will add more citations about software protection and clarified it in the related works.
>
>       [2] Protecting software through obfuscation: Can it keep pace with progress in code analysis? ACM Computing Surveys (CSUR) 49.1 (2016): 1-37
>
>       [3] Protection of Computer Software: Its Technology and Application. Cambridge University Press, 1992.
>
> 3. “Models can be encrypted completely by utilizing TEEs.”
>
> It is true that some edge devices can use TEEs to protect the model. However, TEEs require additional specific hardware support and may significantly increase the latency of the model [4]. Moreover, to use TEEs, a software or model must be designed to support TEEs. In contrast, ModelObfuscator is a general technique that can be applied to almost all mobile devices without requiring any hardware supports or causing any overhead.
>
>
>       [4] Trusted Execution Environment: What It is, and What It is Not, 2015 IEEE Trustcom/BigDataSE/ISPA.
>
> 4. “Clarify the threat model that the reconstruction attacker operates in.”
>
> Thanks for your suggestion.
>
> Threat model: The on-device model is usually saved as a separate file (e.g., .tflite file) and packed into the app package. The attackers can either download the target app from the app markets (e.g., Google Play and iOS App store) or extract the app package file (e.g., APK file for Android, and IPA file for iOS) from hosting devices. These app package files can then be decompiled by off-the-shelf reverse-engineering tools (e.g., Apktool [1] and IDA Pro [2]) to get the original DL model file. Although many on-device DL platforms do not support some advanced functions like backpropagation, attackers can assemble the model architecture and weights into a differentiable model format, or they can use software analysis methods to generate attacks for the target model file [7, 8].
>
> In this paper, we focus on obfuscating the information of the model to disable the software analysis and model conversion tools from extracting model’s information to conduct white box attack.
> We will clarify the threat model in the Methodology section and provide a comparison of three kinds of model extraction attacks.
>
>       [5] https://ibotpeaches.github.io/Apktool/
>
>       [6] https://hex-rays.com/ida-pro/
>
>       [7] Smart app attack: Hacking deep learning models in android apps. IEEE Transactions on Information Forensics and Security, 2022.
>
>       [8] Deeppayload: Black-box backdoor attack on deep learning models through neural payload injection. In 2021 IEEE/ACM 43rd International Conference on Software Engineering (ICSE), 2021.

---

### Decision · Program_Chairs · 2023-01-20

**Decision:**

Reject

**Justification For Why Not Higher Score:**

Please see above

**Justification For Why Not Lower Score:**

NA

**Metareview: Summary, Strengths And Weaknesses:**

The paper presents prototype tool, termed ModelObfuscator, to automatically obfuscate on-device TFLite models. Experiments show that ModelObfuscator increasing the overhead of extracting information about the models (such as structure, parameters and attributes), without increasing latency.

Reviewers raised several concerns:

-The approach relies on "security by obscurity", which is strongly discouraged by many security experts, since an adversary with enough time and resources will be able to extract the valuable information eventually.

-Experiments are not conducted with state of the art model extraction attacks.

-An attacker could simply train a surrogate model, via "model theft", to circumvent the proposed obfuscation strategy.This would result in a black-box attack which is true and would indeed be more costly. However, this does not change the fact that this model would be able to be attacked and extracted, which would finally render the proposed defense useless.



**Summary Of Ac-Reviewer Meeting:**

NA